# One-shot generalization in humans revealed through a drawing task

**Henning Tiedemann[1]\*, Yaniv Morgenstern[1,2], Filipp Schmidt[1,3], Roland W Fleming[1,3]**

[1]Department of Experimental Psychology, Justus Liebig University Giessen, Giessen, Germany; [2]Laboratory of Experimental Psychology, University of Leuven (KU Leuven), Leuven, Belgium; [3]Center for Mind, Brain and Behavior (CMBB), University of Marburg and Justus Liebig University Giessen, Giessen, Germany

**Abstract** Humans have the amazing ability to learn new visual concepts from just a single exemplar. How we achieve this remains mysterious. State-of-the-art theories suggest observers rely on internal 'generative models', which not only describe observed objects, but can also synthesize novel variations. However, compelling evidence for generative models in human one-shot learning remains sparse. In most studies, participants merely compare candidate objects created by the experimenters, rather than generating their own ideas. Here, we overcame this key limitation by presenting participants with 2D 'Exemplar' shapes and asking them to draw their own 'Variations' belonging to the same class. The drawings reveal that participants inferred—and synthesized— genuine novel categories that were far more varied than mere copies. Yet, there was striking agreement between participants about which shape features were most distinctive, and these tended to be preserved in the drawn Variations. Indeed, swapping distinctive parts caused objects to swap apparent category. Our findings suggest that internal generative models are key to how humans generalize from single exemplars. When observers see a novel object for the first time, they identify its most distinctive features and infer a generative model of its shape, allowing them to mentally synthesize plausible variants.

## Editor's evaluation

This paper employs innovative approaches to elegantly tackle the question of how we are able to learn an object category with just a single example, and what features we use to distinguish that category. Through a collection of rigorous experiments and analytical methods, the paper demonstrates people's impressive abilities at rapid category learning and highlights the important role of distinctive features for determining category membership. This paper and its approach will be of interest to those who study learning, memory, and perception, while also contributing to a growing field which uses naturalistic drawing as a window into high-level cognition.

## Introduction

Visual recognition and categorization of objects are vital for practically every visual task, from identifying food to locating potential predators. Humans can rapidly classify objects (*Thorpe et al., 1996*; *Mack and Palmeri, 2015*; *Serre et al., 2007*), complex scenes (*Wilder et al., 2018*; *Fei-Fei et al., 2007*), and materials (*Sharan et al., 2010*) into familiar classes, as well as build new classes when presented with sufficient examples (*Op de Beeck and Baker, 2010*; *Gauthier et al., 1998*). When large numbers of examples are provided, the mathematical basis of human pattern recognition and categorization is well described, and computational models trained with large training sets can

**\*For correspondence:** henning-tiedemann@hotmail.de

**Competing interest:** The authors declare that no competing interests exist.

emulate many of the human abilities in these areas (*Huang et al., 2017*; *Krizhevsky et al., 2012*; *Szegedy et al., 2017*; *Szegedy et al., 2015*; *Szegedy et al., 2016*; *He et al., 2016*; *Radford, 2021*; *Kubilius et al., 2016*; *Jozwik et al., 2018*; *Jozwik et al., 2016*; *Jozwik et al., 2019*). However, humans also have the amazing ability to infer new classes when presented with only a small number or even just one single example, as typically occurs when first encountering new concepts ('one-shot categorization'; *Feldman, 1992*; *Feldman, 1997*; *Ons and Wagemans, 2012*; *Richards et al., 1992*; *Morgenstern et al., 2019*; *Ayzenberg and Lourenco, 2021*). Such one-shot learning—which we here use interchangeably with one-shot categorization—is crucial to the formation of human perceptual categories, particularly at early stages of visual development (*Gelman and Markman, 1986*; *Gelman and Meyer, 2011*; *Gopnik and Sobel, 2000*; *Smith and Slone, 2017*; *Yuan et al., 2020*; *Gershkoff-Stowe et al., 1997*; *Landau et al., 1988*; *Landau et al., 1998*; *Pereira and Smith, 2009*). Yet how we achieve this remains mysterious and is a significant challenge for artificial learning systems (*Geirhos et al., 2018*; *Zhang et al., 2019*; *Zhang et al., 2021*; *Baker et al., 2018*; *Michaelis et al., 2020*).

From a computational perspective, the ability to infer a new category from just a single exemplar seems virtually impossible: Given only a single exemplar of a category, there is an infinite number of sets containing that exemplar, any of which could in principle be the true category from which the exemplar was drawn. How can we predict the scope of a category without having witnessed any variability?

The state-of-the-art in psychology and computer science for understanding how humans generalize from few samples suggest that they infer a *generative model*, which considers the observed exemplar as a single sample from a statistical generative process (*Feldman, 1992*; *Feldman, 1997*; *Fei-Fei et al., 2006*; *Goodman et al., 2008a*; *Goodman et al., 2008b*; *Lake et al., 2015*; *Stuhlmuller et al., 2010*). In intuitive terms, it is assumed that observers have a 'deep' understanding of objects, such that they infer the causes or processes that generated the object (*Fleming and Schmidt, 2019*; *Schmidt et al., 2019*). Importantly, generative models can not only be used to identify behaviourally significant features to judge new samples but can also be used to synthesize or imagine new (i.e., never observed) objects from the same class.

A key step in the inference of a generative model for a given exemplar is the identification of *diagnostic features* that are informative about the underlying generative processes and therefore category membership. Some features—such as the bilaterally symmetric arrangement of animal limbs—are evidence of lawful processes that structure and describe valid members of the category. At the same time, other features of the observed exemplar—such as the specific pose of the limbs— are free to vary across class members. Differentiating between these generic and non-generic (or 'non-transverse'), diagnostic features, such as particular relationships between elements in objects (e.g., symmetry) is one important cue to underlying generative processes (*Feldman, 1997*). Other cue features might include statistical outliers in the shape (e.g., a sharp protrusion, or an angle), that makes a local feature stand out within the shape (*Feldman and Singh, 2005*; *Feldman and Singh, 2006*; *Feldman, 2013*; *Op de Beeck et al., 2008*; *Kayaert et al., 2005*), or compared to others seen previously. However, how we identify and interpret such category-defining features and use them for generalization remains a matter of debate (*Serre, 2016*).

A stumbling block in testing whether humans infer generative models of objects lies in current experimental methods in category learning. Typically, tasks exploring categorization and generalization ask observers to *discriminate* between multiple presented objects (*Ashby and Maddox, 2005*) (with rare exceptions, e.g., *Stuhlmuller et al., 2010*) often varying along binary dimensions (e.g., thin vs. thick, square vs. circle; although see *Hegdé et al., 2008*; *Kromrey et al., 2010*). Yet, the very process of presenting multiple objects potentially interferes with how observers perform the task. Rather than using a generative model, observers can solve these types of discrimination tasks simply by comparing how similar objects are to one another, without establishing preferences for particular features, and without synthesizing new variants. Another key limitation is that the experimenter— rather than the observer—determines the range of possible options the observers can choose from, thereby crucially constraining the range of responses. Thus, a question of central theoretical importance about whether humans use internal generative models— and if so, how consistent they are across observers—remains unanswered.

To overcome these shortcomings by tapping into generative rather than discriminative processes, we used a task in which observers were presented with single Exemplar objects and were asked to

explicitly generate (draw) new objects ('Variations') from the same category, on a tablet computer. With only one Exemplar present, participants could not derive category-defining features by looking for commonalities between category members. Indeed, as their task did not involve comparing objects at all, they could not rely solely on internal discriminative models. Instead, to create new objects, participants were forced to utilize a generative model extracted from the Exemplar, to derive new category members, unless they chose to simply copy the Exemplar with minor deviations, which was discouraged in the instructions. By analyzing the drawings relative to copies of Exemplars, and by asking other participants to (1) categorize drawings, (2) identify their distinctive features, and (3) compare them with Exemplar shapes, we test whether the participants that drew the Variations truly generalized from single exemplars, and determine which features they relied on to do so, giving us insight into what these generative models look like and how similar they are between observers. The strength of unconstrained drawing tasks has been shown in areas such as memory (*Bainbridge et al., 2019*), recognition (*Fan et al., 2020*), efficient representation of both scenes (*Sheng et al., 2021*) and part structure of objects (*Mukherjee et al., 2019*), and developmental changes in children (*Long et al., 2019*), making it an ideal tool to investigate categorization and one-shot learning.

## Results and discussion

### Systematic generalization in a generative one-shot categorization task (Experiments 1 and 2)

Our first major finding is that participants can synthesize categories of complex objects by generalizing from single exemplars. On each trial, one of eight Exemplar shapes (*Figure 1a*) was presented in the upper half of a tablet computer's screen. Participants were instructed to 'draw a novel object that is not a copy of the Exemplar, yet which belongs to the same category' using a digital pencil (**Experiment 1**, *Figure 1b*; see Materials and methods). Exemplars were created by the experimenters by combining a main body with varying numbers of parts (ranging from 0 to 5) and displaying a wide range of visual features, such as positive and negative parts (e.g., the indentation in Exemplar 2), curvatures (polygonal and curved shapes), and complexity of parts (e.g., the 'spike' in Exemplar 4 vs. the 'twirl' in Exemplar 6). The aim was to create Exemplars that are diverse without resembling real-world objects, to maximize generalizations as well as limit the impact of semantic knowledge. While hand-crafting stimuli runs the risk of introducing biases, the intended variety of Exemplars made an algorithmic creation approach unfeasible. For each of the 8 Exemplars, 17 participants ('drawers') each drew 12 Variations (yielding a total of 204 Variations per Exemplar, i.e., 1632 drawings overall). To minimize potential carry-over effects of previously seen Exemplars, shapes were shown in randomized order for each participant. As a baseline, another group of participants ($n$ = 15) was asked to copy the Exemplars three times as accurately as possible (**Experiment 2b**).

To assess the range and variety of the generated shapes, a new group of 12 participants rated the similarity of each Variation to the corresponding Exemplar (**Experiment 2**, note that number of data points per participant vary by experiment, for details see Materials and methods). *Figure 1c* shows examples of Variations for one Exemplar, with perceptually more similar drawings plotted closer to the Exemplar (see *Figure 1—figure supplement 1* for Variations of other Exemplars). The drawings illustrate the range of responses, with some shapes being very similar to the Exemplar while others differ considerably. *Figure 1d* shows the average similarity of produced Variations for each participant, demonstrating that participants varied substantially in how much their drawings deviated from the Exemplar, that is, in their creativity when producing Variations. Linear regression showed that age of drawing (i.e., whether a shape was drawn earlier or later within the sequence of 12 shapes for each Exemplar), did not predict similarity ($R^2 < 0.01$, $F(1,10) = 0.08$, p = 0.77). Individual participants tended to stay within a constrained similarity spectrum with only small trends in the direction of less or more similar shapes over time with slopes ranging from −0.13 to 0.12 (mean = 0.07).

To test whether the generated shapes were genuinely new objects or just slightly altered copies of the Exemplars, the copies were compared to a subset of Variations using a 2-AFC task (**Experiment 2b**, see Materials and methods for details), in which a new group of participants ($n$ = 15) was asked to pick the shape that looked more like a copy of the Exemplar. In 95% of trials (chance = 50%) the copies were chosen, showing that the vast majority of a representative subset of Variations were perceived

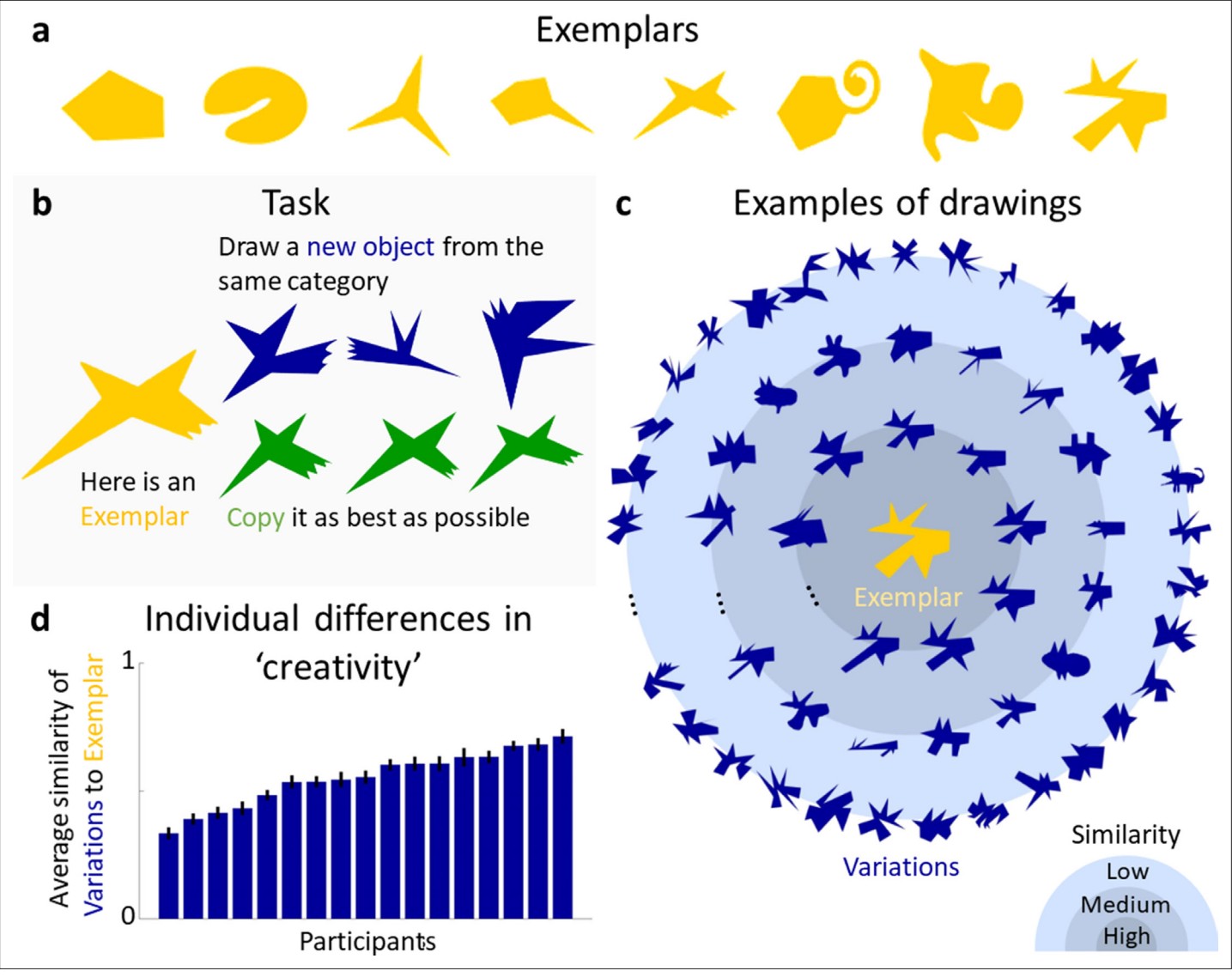

**Figure 1.** Generative one-shot categorization task and results. (**a**) Exemplars presented in the Experiments. (**b**) In the task, a group of participants was presented with an Exemplar and asked to draw new objects from the same category (blue). As a measure of baseline performance, another group of participants was asked to copy the Exemplar as accurately as possible (green). (**c**) Examples of drawn Variations (blue) generated in response to one of the Exemplars (yellow). Variations are plotted according to their perceptual similarity to the Exemplar, with more similar Variations closer to the centre. (**d**) Individual differences in 'creativity', defined by the average perceived similarity of participant's drawings to the respective Exemplar. Error bars indicate standard errors.

The online version of this article includes the following figure supplement(s) for figure 1:

**Figure supplement 1.** A subset of Variations (dark blue) created for each Exemplar (yellow, centre).

to be more different from the Exemplars than a mere copy (one-sided binomial test: 50%; $N$ = 16,200, i.e., number of judgements; $K$ (correct responses) = 15,388, p <0.001).

## Responses represent distinct perceptual categories (Experiment 3)

Next, we tested whether the drawings represented distinct perceptual object categories or were mere random variations that did not form coherent groups. The similarity ratings for each Exemplar's Variations from **Experiment 2** were split into subsets (40 bins) that spanned the full similarity continuum (see Materials and methods). A new group of participants ($n$ = 15) classified one Variation from each bin for each Exemplar (320 stimuli in total), by sorting in each trial a randomly selected Variation into one of the eight Exemplars' categories. The average percentage of correct

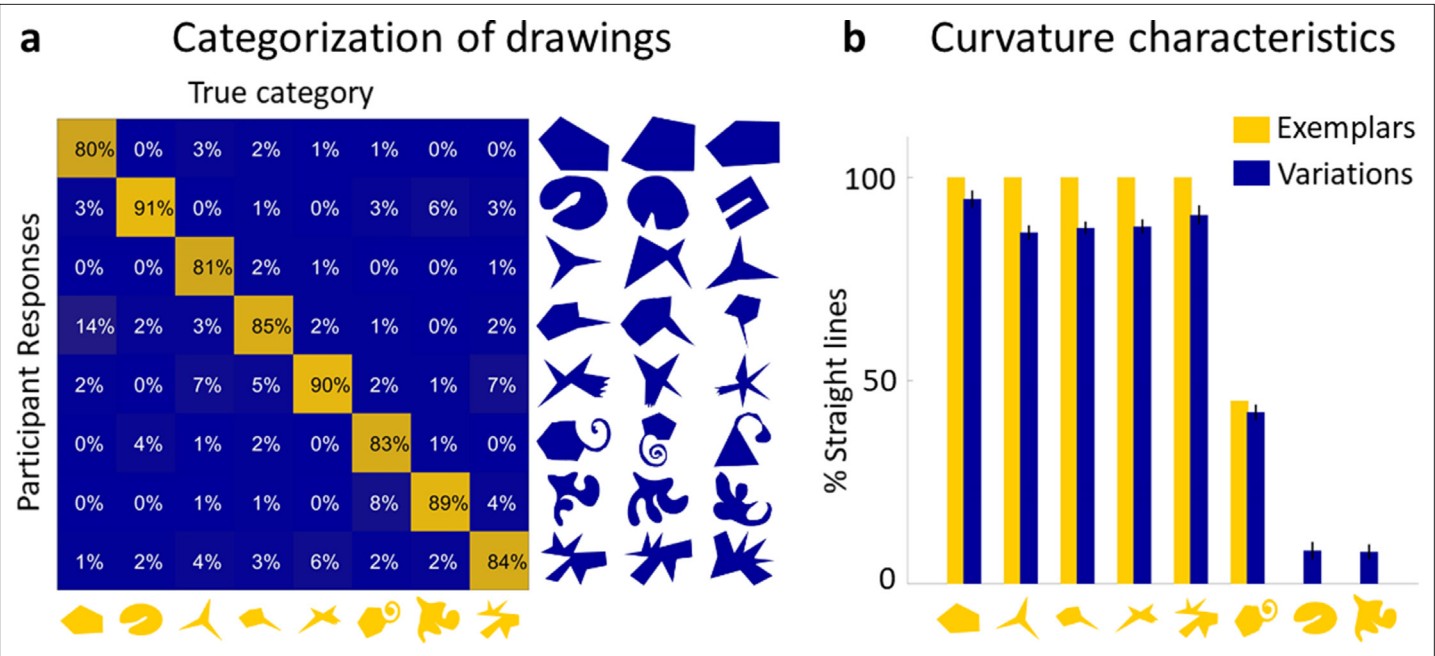

**Figure 2.** Drawings constitute a real perceptual category. (**a**) The confusion matrix for true classes and participant responses shows that the vast majority of Variations were classified correctly. A subset of Variations that had to be categorized is shown on the right of the matrix with the Exemplars on the bottom. (**b**) Variations largely reproduce the global curvature characteristics of the Exemplar. Curvature similarity across Variations and their Exemplars: Exemplars (yellow) ordered by percentage of perimeter comprised of straight lines, together with average percentage of straight lines in all Variations of that category (blue). Error bars indicate standard error.

classifications was high (86%) and well above chance (one-sided binomial test: 12.5%; $N$ = 4800, i.e., number of judgements; $K$ (correct responses) = 4111, p < 0.001). *Figure 2a* shows the confusion matrix for true classes and participant responses. In almost all instances, observers sorted the Variations correctly into the originating class. Except for a single cell (row 4, column 1), all cells are significantly different from chance with all diagonal cells above—and all others below—chance (one-sided binomial tests with Bonferroni-adjusted p value of 0.015, according to the eight possible outcomes in each row). To investigate whether performance in this task was influenced by the similarity between Variations and Exemplars, the Variations were divided into four similarity bins ranging from very similar to very dissimilar. Performance in the three bins most similar to the Exemplars was virtually identical (overall 89%, 87%, and 88% correct responses), with a maximum of 14% misclassifications in single cells. For the least similar bin, performance dropped to 79%, suggesting that highly dissimilar shapes were significantly more often misclassified, while overall classification accuracy was still way above chance. The maximum of misclassification here was 28% in a single cell (row 4, column 1). This shows that there are almost no systematic miscategorizations between Exemplar categories and demonstrates that the Variations produced in the generative task are samples of robust perceptual categories. Specifically, our results tend to suggest that drawers identified diagnostic features in the Exemplars and reproduced them in the Variations, allowing other observers to identify them as belonging to a common class. Thus, drawers effectively derived distinct novel categories from just a single object.

## Identifying category-defining features

So far, we have shown that drawers generated genuinely novel objects, which other observers could nevertheless assign to their corresponding category. This suggests that drawers identified and reproduced those features of the object that were most diagnostic of the class. Thus, we next investigated which features were preserved, the extent participants agreed about the most significant features, and the importance of these features for category assignment.

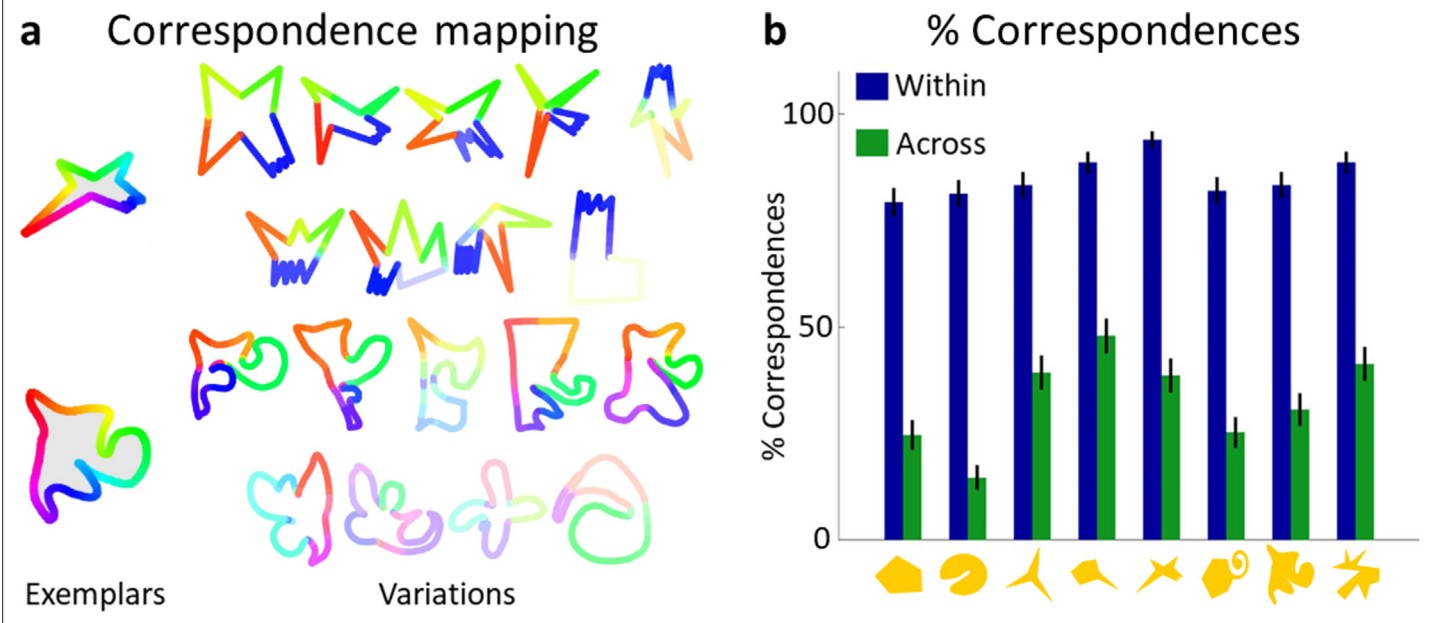

**Figure 3.** Variations and Exemplars of the same category share most of their parts. (**a**) Aggregated mapping of correspondences for two categories, showing high agreement between participants about corresponding parts. Colours are explained in main text. (**b**) Comparison of percentage of correspondences seen within (blue) and across (green) categories, showing that significantly more correspondence was perceived within categories. Error bars indicate standard error.

The online version of this article includes the following figure supplement(s) for figure 3:

**Figure supplement 1.** Correspondence data for whole subset.

**Figure supplement 2.** Analysing part order and part changes in the correspondence task (Experiment 4).

## Global curvature features

One significant feature preserved in almost all Variations was the global curvature, namely whether the Exemplar consisted of straight or curved lines. We find that polygonal Exemplars tended to lead to polygonal Variations while curvaceous Exemplars led to curvaceous Variations ($r = 0.99$, p < 0.001; *Figure 2b*). This finding is broadly in line with the concept of non-random features indicating generative processes (*Feldman, 1992*; *Feldman, 1997*): A pencil tracing a random walk is unlikely to draw a straight line, so straight lines are considered evidence of a significant (i.e., non-random) generative process, which are therefore preserved in Variations.

## Part-related features (Experiment 4)

Considering the Variations in *Figure 1c*, another significant feature that seems to be approximately preserved in many drawings is the part structure—that is, arrangement and number of parts—of the Exemplars (this is especially salient when looking at the most similar Variations; see also **Ideas and Speculations**). At the same time, the parts themselves were often modified in size, orientation, or elongation. This signifies a potential strategy that participants might have used in the drawing task, highlighting their generative approach: Starting from perceptual segmentation of the Exemplar into parts, they may have modified these parts (to varying degrees) and 'put them back together', either in the original or a different order.

   To test whether participants used this strategy, another group of participants ($n = 15$) was shown one Exemplar at a time, paired with a Variation of either the same or a different category, and asked to identify any corresponding parts between the two shapes. This allowed us to test whether part correspondence is stronger within categories than across categories—and how much of the Exemplar's part structure was retained in its Variations. When participants perceived parts as corresponding, they delineated each part by drawing a line, thereby 'cutting' the shape into two parts on either side of the line, and then choosing one of the parts to indicate the correspondence to a part in the Exemplar.

*Figure 3a* shows aggregated mappings of correspondence for two of the categories, with the Exemplar on the left and rows of corresponding Variations in descending similarity to the Exemplar (see *Figure 3—figure supplement 1b* for complete set). Each point of the Variations' contour is coloured the same as the Exemplar's contour to which it was perceived to correspond the most. If a point was perceived to correspond to a whole section of the Exemplar (e.g., the green 'nose' of the Exemplar in the bottom row of *Figure 3a*), then it was coloured the same as the circular median point of that section. Colour saturation indicates how often correspondence was seen for each point, with higher saturation indicating stronger correspondence. This visualization shows the generally high level of agreement about corresponding parts between participants. *Figure 3b* compares the correspondence in same-category pairs to that in different-category pairs. On average, 71% of an Exemplar's area corresponded to some part of its Variations, with only 18% correspondence to Variations of other categories (one-sided *t*-test: $t(1199) = 38.84$, p < 0.001, *Figure 3—figure supplement 1a*). To test whether less similar shapes share fewer parts, we analyzed the corresponding area as a function of the similarity of Variations to Exemplars by linear regression. We find that the extent to which the Variation was seen to correspond was well predicted by similarity ($R^2 = 0.92$, $F(1,8) = 96.69$, p < 0.001, see *Figure 3—figure supplement 1a*). This also suggests that the extent to which parts or area corresponded might have been used by observers in **Experiment 2** to assess similarity.

A significant correlation between the percentage of correct categorizations from **Experiment 3** and the percentage of corresponding area ($r = 0.53$, p < 0.01) indicates that Variations that share more parts with the Exemplar are grouped more often with their Exemplar compared to those sharing fewer parts. Notably, out of the 80 Variations used in **Experiment 4**, 49 were always classified correctly, with a range in corresponding area between 30% and 100%, and a mean of 81%.

Visual inspection of Variations suggested that Exemplars' part orderings were often retained in Variations, that is, visually different corresponding parts were arranged in the same sequence. However, in some Variations this order was changed, as if the parts were 'shuffled'. To test the prevalence of different part orderings in the Variations, we created a compressed representation of part order for each shape pair's correspondences (*Figure 3—figure supplement 2*): Moving clockwise around the shape's silhouette, we listed each part indicated by the participant, as well as gaps with no correspondences, resulting in a 'part circle' describing the order of labelled parts for each shape (see *Figure 3—figure supplement 2a*). This representation was chosen because Exemplars were created to comprise a main body with parts arranged roughly circularly around it, making the resulting part circles approximately analogous to perceived part ordering. To analyze part order, part circles were then compared to quantify the extent to which orders were identical, reversed, or shuffled (i.e., non-identical). There is an important limitation to this analysis, however. By this definition, part order could only be changed with more than two corresponding parts, amounting to 30% of all trials. For the remaining 70% of trials, this analysis is not possible as with two or fewer corresponding parts, part order is automatically conserved or cannot be defined. This relatively low number is explained by participants often aggregating corresponding parts in their responses, for example, by considering the three 'non-bitten' spikes in the top Exemplar in *Figure 3a* as a single corresponding part—resulting in only two correspondences even though the shape may have featured more perceptual parts. Of those trials with more than two correspondences, part order was identical in 77%, reversed in 7%, and shuffled in 17% of cases. Analyzing gaps in part circles also provided a tentative measure of how often parts of the Exemplar were omitted (4%), substituted with another non-corresponding part (25%), or new parts were added that did not exist in the Exemplar (8%). Notably, more part substitutions occurred for less similar Variations to the Exemplar (*Figure 3—figure supplement 2d*). For further details of this analysis, see Materials and methods.

While we have to be careful to generalize to the full set of Variations from this limited number of data points, this analysis nevertheless provides some insight into potential strategies used in the creation of new shapes: Many Variations retained the Exemplar's part order, sometimes in reversed form, suggesting that the shape was changed from its original form (as opposed to being created completely from scratch) either globally or part by part to create an appreciably different object displaying the same part order. In contrast, Variations with a shuffled part order, or omitted and added parts, point to a 'building block' approach where parts are treated as independent, re-combinable elements.

Together, these findings demonstrate that Exemplars and associated Variations share a considerable portion of their parts, indicating that drawers preserve identifiable parts in their Variations even though they varied specific geometrical properties of these parts. Furthermore, Variations also tended to preserve the order of corresponding parts—with some exceptions where part order was changed—overall pointing to a highly part-based creation approach.

## Identification and preservation of distinctive parts (Experiment 5)

Another notable feature of the generated shape categories is that certain parts of the Exemplars seem to be more distinctive than others, and that these often also appear in the Variations. We suggest that these distinctive parts are a major driving force for correct categorization (**Experiment 3**). To address whether participants agreed about which shape features are most distinctive, we showed a new group of participants ($n$ = 10) all 8 Exemplars together with 39 Variations from each of the categories, in random order. They were asked to mark up to three parts of each object's silhouette through a painting interface, starting with the most distinctive part, followed by the second and third most distinctive parts. *Figure 4a* shows all responses for one shape, together with the aggregated response, which indicates a high level of agreement across observers (see *Figure 4—figure supplement 1* for the complete set). *Figure 4b* shows aggregated responses for a subset of Variations per category. Contrasting these data with randomized responses mimicking the number of areas painted and the lengths of consecutive areas painted, but with randomized placement of those areas (see Methods for details), shows significantly higher agreement between humans compared to chance: The mean consecutive area of a shapes' silhouette with a high distinctiveness score (above 75% of the highest score) for the human data made up 19% of a shape's perimeter compared to <1% for the randomized data (two-sample Kolmogorov–Smirnov test, p < 0.001). This high agreement among observers is especially noteworthy given that the quite vague concept of 'distinctiveness' could be interpreted differently by different observers, pointing to just how important and characteristic these parts are considered compared to the rest of the shape.

Visual inspection suggests that for most categories, participants tend to consistently indicate specific parts as being the most distinctive across different Variations (e.g., indentation for category 2, spike for category 4, jagged feature for category 5, and twirl for category 6). Cross-referencing these data with the part correspondences from **Experiment 4** allowed us to test whether distinctive parts within a category are indeed conserved (i.e., whether the most distinctive parts in each shape are those that are judged to correspond to each other). For each corresponding part pair, average distinctiveness scores of both parts were calculated. Comparing the scores of all Exemplar parts with the scores of all Variation parts of the same category shows a high correlation ($r$ = 0.63; p < 0.001), suggesting that distinctive parts remain distinctive even when modified or shuffled within the overall part structure. Equivalently, indistinctive parts in Exemplars tend to remain indistinctive in Variations, lending further support to the finding that participants modify individual parts to create new shapes. This raises the possibility that parts that are perceived to be distinctive are particularly important in determining category membership.

## Causal role of distinctive parts in categorization (Experiment 6)

We next sought to test the impact of distinctive parts on category membership more directly. To do this, we created new stimuli from Variations by replacing the most distinctive part—as determined from **Experiment 5**—with the most distinctive part of the Exemplar of another category. For comparison, we created stimuli where we replaced the same Variation's least distinctive part with the least distinctive part of the Exemplar of another category (*Figure 5a*; controlling for perimeter length—see Materials and methods). A new group of participants ($n$ = 15) grouped each of these newly generated stimuli (280 with the distinctive, 280 with the indistinctive part swapped) with one of the Exemplars, similar to **Experiment 2**.

*Figure 5b* compares the results with the percentage of correct categorizations in **Experiment 2**. Binomial tests indicate that the proportion of correct responses from **Experiment 2** (86%) was significantly higher than both swap conditions (69% for indistinctive swap, p < 0.001, and 32% for distinctive swap, p < 0.001). Further, we find a large difference in effect sizes: while swapping an indistinctive part does have a small effect (Cohen's $h$ = 0.29), swapping a distinctive part has an immense effect on correct categorizations (Cohen's $h$ = 1.05). This shows that distinctive parts were a driving force

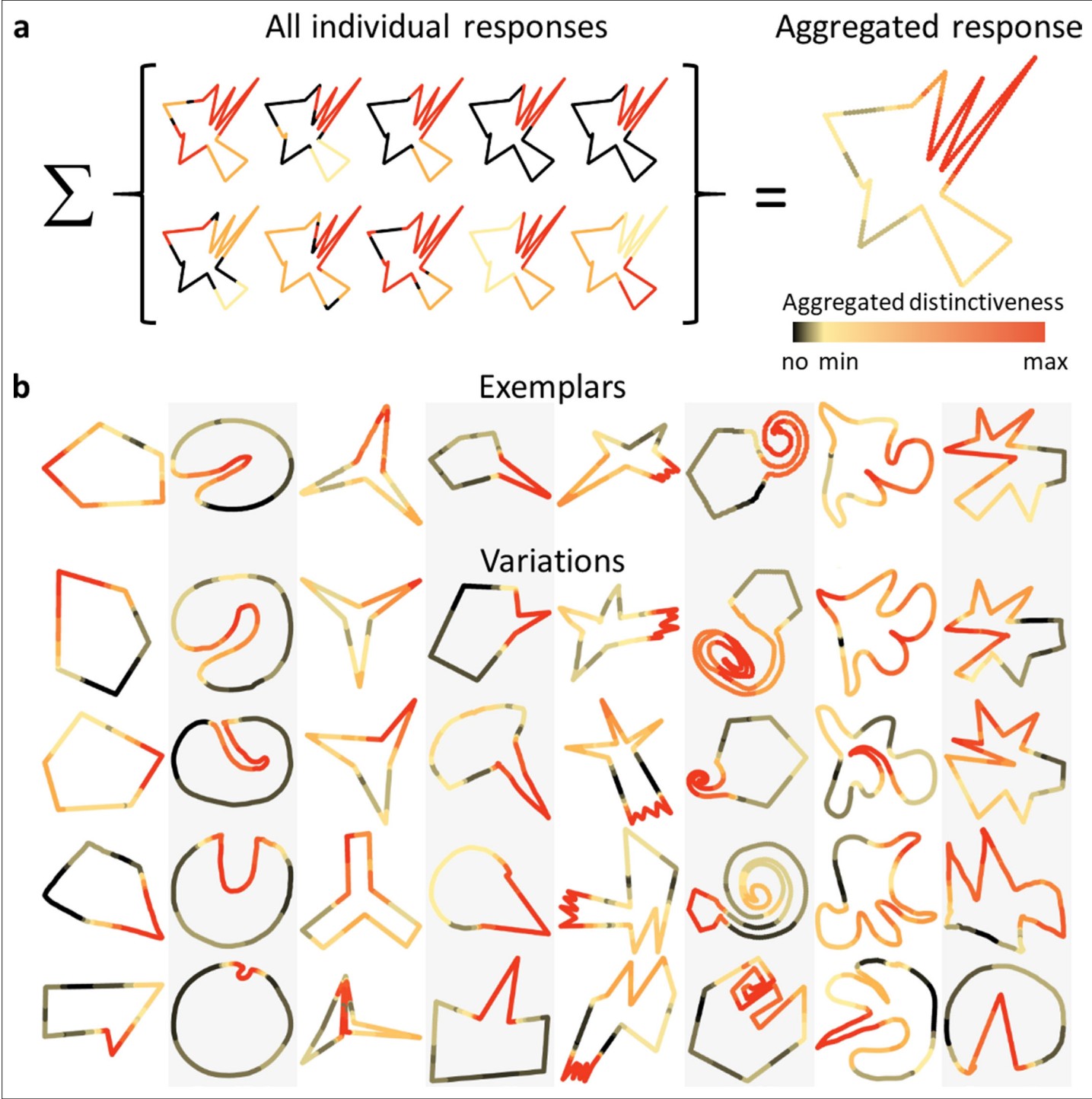

**Figure 4.** Observers agree on the most distinctive parts. (**a**) Individual responses of all participants for an example Variation, showing which parts were marked as most distinctive (red), second most distinctive (orange), and third most distinctive (yellow). Aggregating these responses results in the shape on the right. The redder each point in the contour, the more distinctive it was evaluated across all participants. (**b**) Comparing aggregated responses between Exemplars and corresponding Variations (a subset shown here) suggests that in most categories similar parts (e.g., the indentation, spike or twirl) were identified as distinctive across most Variations.

The online version of this article includes the following figure supplement(s) for figure 4:

**Figure supplement 1.** Aggregated distinctiveness scores for all shapes.

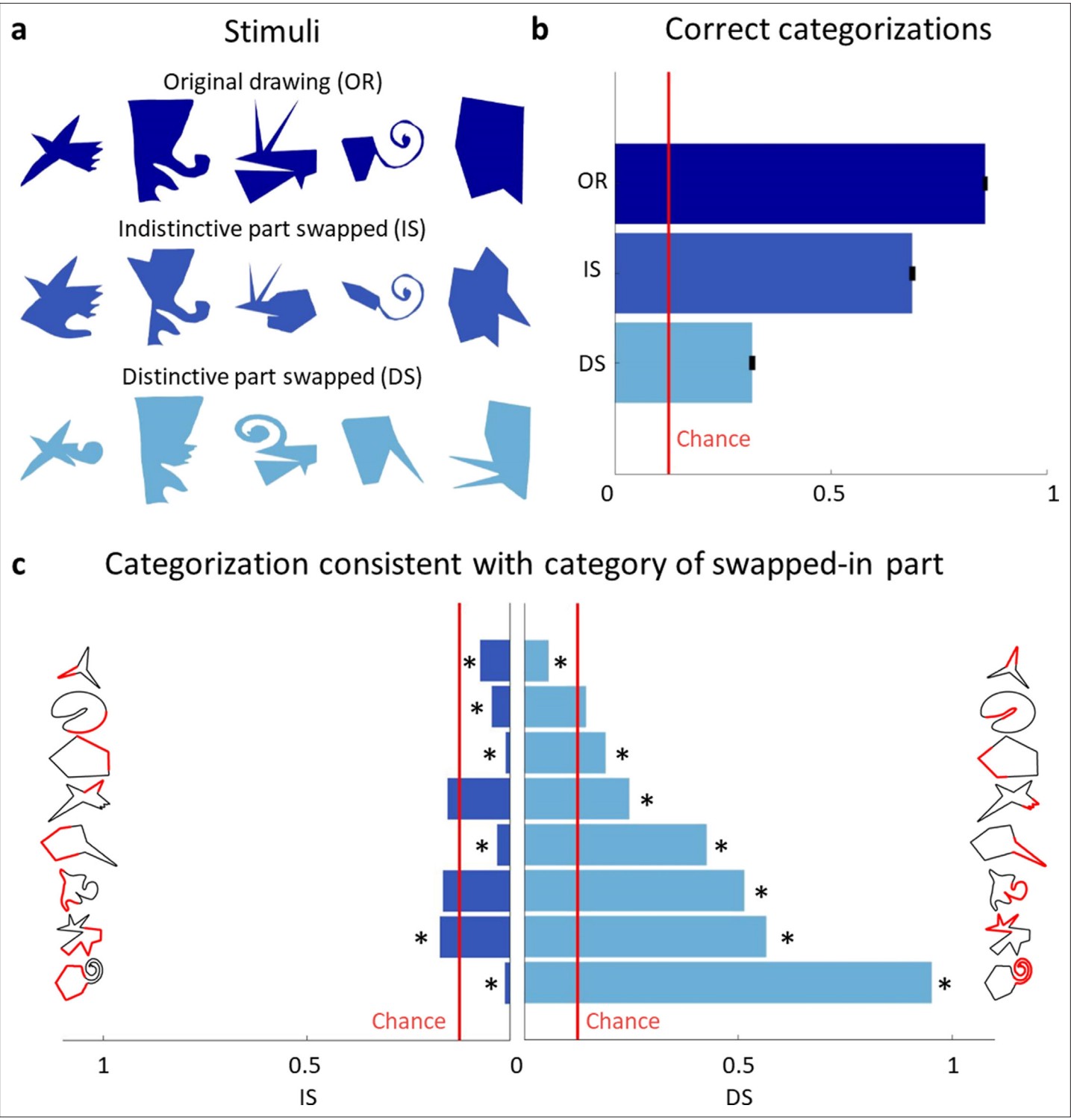

**Figure 5.** Distinctive parts are the main categorization cue. (**a**) Original drawings (top row) were altered so that either the least distinctive part was swapped with the least distinctive part of a different category (second row) or the most distinctive part was swapped with the most distinctive part of another category (third row). (**b**) Comparison of the percentage of correct categorizations for the original shapes (from **Experiment 2**), swapped indistinctive parts and swapped distinctive parts (error bars indicate standard errors). (**c**) Bar plots showing how often the category of the swapped-in part determined the categorization choice. The indistinctive (left) and distinctive (right) parts are shown in red on the shapes' silhouette. Bars significantly different from chance (p < 0.01) are marked with an asterisk.

in categorization decisions, even though they often comprised only a small percentage of shapes' contours. *Figure 5c* summarizes how often the category of the new, swapped-in part determined the categorization decision, separately for indistinctive and distinctive parts. Performing one-sided binomial tests on each of swapped-part conditions (with a Bonferroni-adjusted p level for 16 tests) shows that most distinctive parts strongly biased participant's responses toward their categories. In contrast, only one indistinctive part had a significantly positive impact on its category choice, while most indistinctive parts resulted in choices of their category even significantly below chance.

Previous studies showed that small-to-intermediate fragments of an image are sufficient for correct categorization (*Hegdé et al., 2008*; *Ullman et al., 2002*). These informative fragments were defined by implicit analysis of the statistical distribution of features across the complete object category or categories (i.e., the fragments were learned during a training phase). In contrast, in our experiments, parts were identified as distinctive from just a single piece of data (i.e., one object), consistently across many observers. We suggest that observers may use statistical outliers within shapes (*McCarthy and Warrington, 1986*), or outliers relative to previously seen objects to identify distinctive parts.

In summary, observers agreed on the most distinctive parts of shapes and used this information as one of the main cues for category membership. In line with this, when creating new shapes, these distinctive parts were reproduced with modifications that retained their specific characteristics making them both distinctive and signifiers of their category.

## General discussion

Our ability to organize objects into categories at a glance is of fundamental importance to everyday activities. It allows us to access a wealth of knowledge about objects from previous experiences, rather than having to discover each newly encountered item's properties de novo. Visual object categorization is typically so effortless that we easily take it for granted. Yet disorders of object perception—such as visual agnosia (*McCarthy and Warrington, 1986*; *Riddoch and Humphreys, 1987*; *Goodale et al., 1991*; *Behrmann and Nishimura, 2010*) —lead to profound deficits. A particularly striking characteristic of healthy human object perception is how rapidly observers learn new categories. Whereas machine-learning systems typically require thousands of examples per category to rival human performance at recognizing objects in photographs (*Huang et al., 2017*; *Krizhevsky et al., 2012*; *Szegedy et al., 2017*; *Szegedy et al., 2015*; *Szegedy et al., 2016*; *He et al., 2016*; *Radford, 2021*; *Kubilius et al., 2016*; *Jozwik et al., 2018*; *Jozwik et al., 2016*; *Jozwik et al., 2019*), human infants and adults appear to be able to generalize successfully from just a single example of a new category (*Gelman and Markman, 1986*; *Gelman and Meyer, 2011*; *Gopnik and Sobel, 2000*; *Smith and Slone, 2017*; *Yuan et al., 2020*; *Gershkoff-Stowe et al., 1997*; *Landau et al., 1988*; *Landau et al., 1998*; *Pereira and Smith, 2009*)—so-called 'one-shot learning' (*Feldman, 1992*; *Feldman, 1997*; *Ons and Wagemans, 2012*; *Richards et al., 1992*; *Morgenstern et al., 2019*; *Ayzenberg and Lourenco, 2021*).

One-shot categorization is a formally under-constrained inference problem (*Feldman, 1992*). There are infinitely many sets containing any given exemplar, any of which could be the true category. It is thus remarkable that humans seem to be able to draw consistent conclusions about category membership from such sparse data (*Feldman, 1992*; *Feldman, 1997*). Their judgements presumably reflect assumptions about how—and by how much—objects within a category tend to differ from one another, which constrains the space of variants that are deemed likely. Yet how this occurs remains elusive.

We suggest that generative statistical models of shape play a key role in one-shot categorization. However, to date, direct evidence for generative models has been scant. Most experimental studies on categorization use sets of pre-selected stimuli to define categories and *discriminative* tasks to test their hypotheses. The main drawback of this approach is that the assumed category-defining features are determined by the experimenter and might therefore differ from the features used in unrestricted categorization decisions. Moreover, discriminative tasks allow for simple strategies based on comparisons between objects, rather than probing the visual system's internal generative models of objects which are thought to be central to how humans and machines can learn categories from sparse data (e.g., *Fleming and Schmidt, 2019*).

Here, by contrast, we used a generative one-shot categorization task to tap directly into generative models and human creativity—and to identify category-defining features—by allowing participants

to generate their own new category samples, rather than merely selecting between experimenter-generated alternatives. The resulting Variations are thus shaped by the features that participants consider important in the context of their previous experiences and by their internal visual imagery processes. In principle, different observers might consider different features as significant so that across observers no coherent object categories would emerge. However, in our findings this was clearly not the case. We found a high degree of agreement between observers, suggesting general principles of how humans analyze single objects and extrapolate new category members from its features. Participants created Variations for different Exemplars varying in curvature characteristics and number of parts, resulting in a large dataset of over 1600 drawings, which were significantly more variable than mere copies of the Exemplars. Yet, despite the wide variety of Variations, the overwhelming majority of a representative subset of these shapes was correctly grouped with the originating Exemplar by independent observers, showing that our task yielded genuine distinct perceptual categories.

There are, of course, some limitations to our approach. By asking participants to draw variants, the responses are necessarily limited by participant's skills with the digital drawing interface. Some people have lesser confidence or ability in drawing, and thus it is possible that some of drawings might not accurately reflect their mental image of a given variant. However, this is somewhat mitigated by the relatively straightforward 2D forms they were asked to draw. No ability to render perspective, shading or other more advanced artistic skills were required. It is also worth noting that there was a substantial difference between copies (which were quite accurate) and variants (which were highly diverse). This suggests that artistic ability did not prevent participants from depicting at least a subset of the varied and novel instances brought to mind by the exemplar.

Another potential bias introduced by our approach is possible carry-over effects of Exemplars seen in previous trials influencing the drawings of later ones. While we aimed to minimize these effects by randomizing the order of Exemplars, future studies could reduce this further by presenting just a single Exemplar per participant, for example in an online experiment with larger numbers of participants.

A third potential issue with the method was our use of hand-created experimental stimuli. The presence of distinctive features in some of the stimuli (e.g., the 'twirl' in Exemplar 6) was therefore not accidental but the result of a decision by the experimenter, and it could be that such features occur less frequently in natural objects. This limitation could be mitigated in future experiments by creating stimuli through a stochastic process that is less under experimenter control. For example, by training a generative algorithm (e.g., Generative Adversarial Networks *Goodfellow et al., 2014*) on a dataset of natural shapes, it is possible to create stimuli that share statistical properties with natural shapes

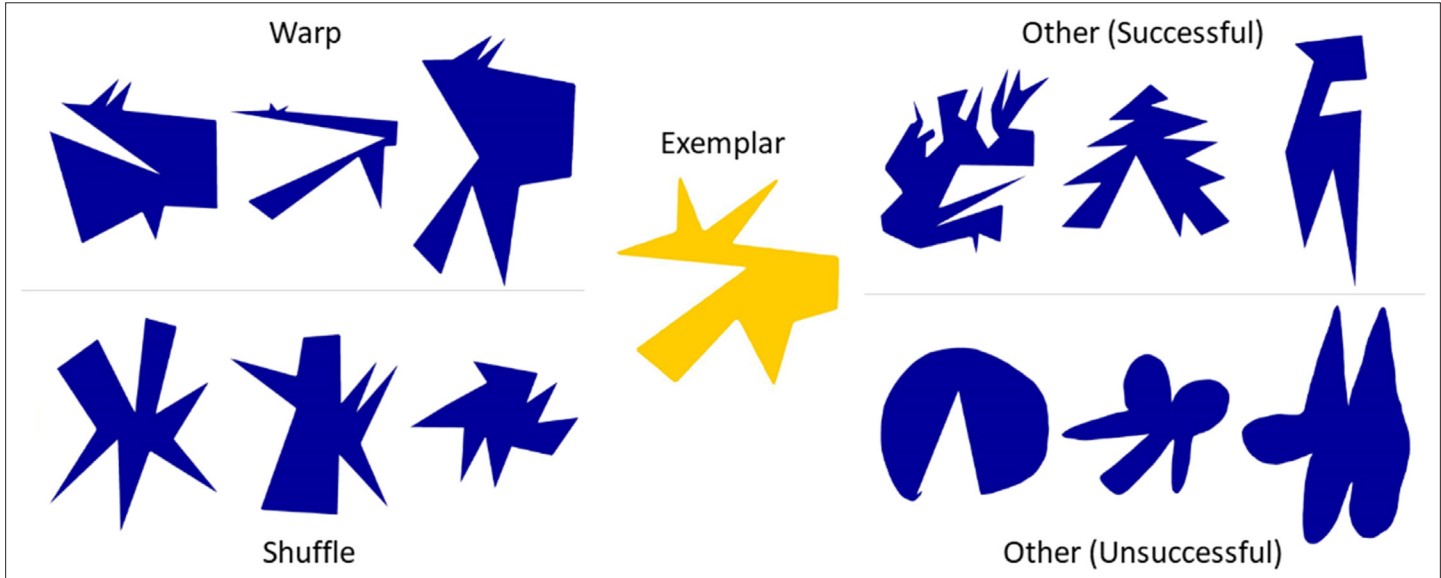

**Figure 6.** Examples of proposed strategies used by participants. Selected examples representing part-based strategies like warp, shuffle, and other strategies, both successful (i.e., shapes were correctly classified almost all the time) and unsuccessful strategies (i.e., shapes were correctly classified far below average).

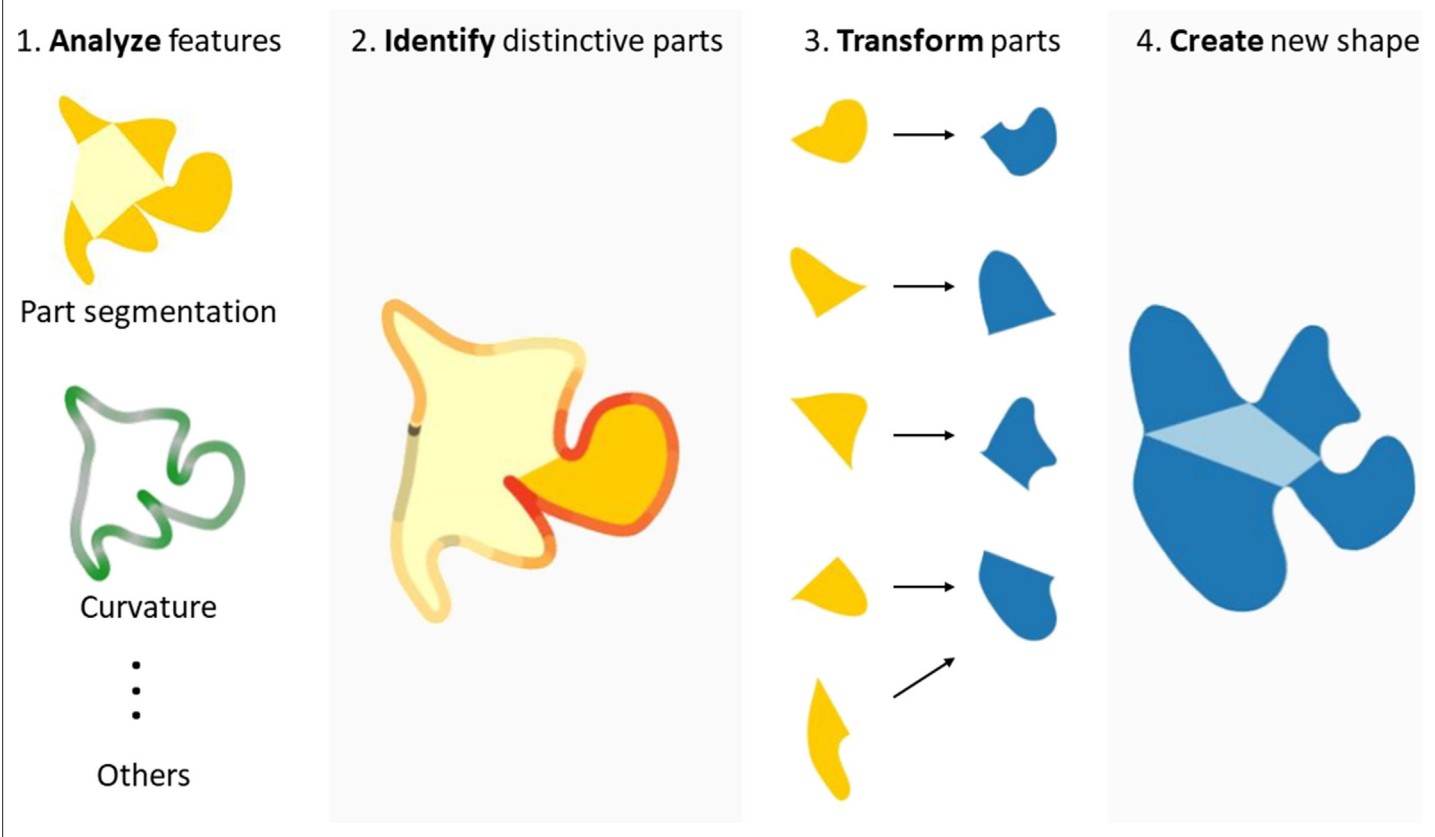

**Figure 7.** Proposed steps of shape creation using a predominantly part-based approach. After analyzing shape features and identifying distinctive parts, the individual parts get transformed and re-assembled to form a new shape. In the transformation process, distinctive parts are changed to a lesser extent than other, non-distinctive parts. The creation of a new shape from the individual parts might involve adding, merging, or removing parts, as well as changing their order. Note that this is an illustration of only one strategy (albeit an important one); however, Variations might also be created using non-part-based strategies, where other features such as curvature are varied.

(*Morgenstern et al., 2021*), although care would need to be taken to rule out stimuli that resemble recognizable items (which often result from such methods). However, even algorithmic stimulus generation does not fully solve the problem. Through their decisions about the algorithm and training set, experimenters necessarily exert some degree of control over the kinds of stimuli presented to participants, even if they do not hand-create or hand-select the individual instances, as we did. Despite our use of hand-drawn Exemplars, the finding that participants could consistently identify distinctive features without any explicit training—and treated these as defining class membership—suggests that both experimenter and participants shared assumptions about which features are important. More broadly, the finding that participants created Variations that other participants recognized as belonging to the same class provides an existence proof for systematic generative processes, irrespective of how frequently such distinctive features occur in natural objects.

## Ideas and speculation

How might a generative model operate in practice to allow a participant to synthesize new objects? While it is difficult to probe the exact strategies used to create each Variation, visual inspection of the drawings, as well as the part correspondence task in **Experiment 4,** provide some tentative insights. We summarize our speculations about some of the possible strategies in *Figure 6*. For example, we call one strategy 'Warping', as many variants appear to be related to the original Exemplar by a relatively straightforward (non-linear) spatial distortion operation that preserves the ordering of points, but alters their relative distances. Through such warping, a wide variety of shapes can be synthesized while largely retaining one-to-one correspondence with features of the prototype from which they are derived.

Some strategies seem to involve segmenting the Exemplar into parts, which are then modified to various degrees and recombined to create a Variation (*Figure 7*). There is considerable evidence for perceptual organization processes that segment shapes into distinct subcomponents based on their geometrical properties (*Hoffman and Richards, 1984*; *Hoffman and Singh, 1997*; *Siddiqi and Kimia, 1995*; *Biederman, 1987*; *Singh et al., 1999*; *Marr and Nishihara, 1978*) and causal origin (*Spröte et al., 2016*). It thus seems intuitively plausible that parts might play a significant role in the generative process too. This idea is further supported by the following observations: (1) Variations' parts often corresponded to parts of the Exemplar (**Experiment 4**); (2) part structure was retained or merely shuffled in a significant number of shapes (**Experiment 4**); and (3) distinctive parts were crucial for categorization (**Experiment 6**). As a result, even highly dissimilar shapes still shared 34% of their area with the originating Exemplar (see *Figure 3—figure supplement 1a* right panel). Importantly, some parts were identified as being highly characteristic of the Exemplar's category and accordingly were varied in the Variations in ways to remain category defining (**Experiment 6**).

In addition to reusing and changing individual parts, the order of parts also seems to have been retained in many Variations—showing that drawers relied on both individual parts as well as their relationships to create a new object. Indeed, even shapes that appear to be created by a straightforward 'Warp' strategy are not inconsistent with a part-based analysis and recombination.

In other Variations, however, part order was shuffled, illustrating a local strategy in which parts were considered independent and recombined in different order. This part-based approach is also reflected by the omission or addition of parts to the Exemplar, as revealed by our 'part circle' analysis. One limitation of the analyses supporting these suggestions is that in **Experiment 4**, observers often (70% of trials) indicated only two or fewer corresponding parts. For these shapes, it is not possible to identify part-order relationships using the 'part circle' analysis. This was in part because participants often appear to have bundled together multiple elements—which a part-detection algorithm would treat as discrete—into composite shape elements, effectively processing the parts in concert. Moreover, although the drawings themselves were entirely unconstrained, it should be noted that **Experiment 4** was explicitly focussed on corresponding parts by design, which only allows us to assess part-based strategies.

Visual inspection of the dataset also suggests additional strategies that are harder to quantify and less decidedly part focussed. Some of these are shown in *Figure 6* (top right), which seem to change the Exemplar's part structure substantially while nevertheless retaining and exaggerating the (statistical) motif of 'spikiness'. The exact nature of the features that define abstract concepts such as 'spikiness'—and thus allow observers to relate the Variations to the Exemplar—remain elusive. However, presumably it is necessary to retain certain statistical properties of the shape, such as the prevalence of sharp angles and straight edges. The bottom right panel shows Variations which were less often linked to the original category in **Experiment 3** than the average Variation. Very few if any of the parts in the Exemplar are retained, and curved elements abound despite the original shape being entirely angular. Evidently, eliminating core features—such as parts or straightness—was not a successful strategy for generating legal Variations. Consequently, we suggest that observers analyze the Exemplar with respect to a number of features, ranging from basic properties like curvature to more high-level properties like 'spikiness', and then vary one or multiple of these features to create a new object. This is consistent with recent work showing that perceived similarity between shapes can be predicted by considering a large number of shape features (*Morgenstern et al., 2021*). Yet, in general, observers seemed to share assumptions about the extent to which features could be varied while still retaining the categories' identity, as indicated by the small number of misclassifications in **Experiment 3**. It is interesting to speculate that these assumptions may therefore be derived from the statistics of variations between items in the natural world.

Many of the generated shapes not only shared most of their parts, but, strikingly, specific parts were also reliably perceived as more distinctive than others and were the main catalyst driving categorization decisions, even if they comprised only a minority of the shapes' area. This suggests that a crucial stage in one-shot categorization is the identification of those features within an object that are most likely to be 'distinctive' for defining the category.

What is the basis of such inferences? Given only a single exemplar, how are the 'most distinctive' features determined? We suggest that observers' decisions are related to processes that identify signatures of the underlying *generative processes* responsible for creating the observed shape and

which thus define the category. This is closely related to Feldman's (*Feldman, 1992*; *Feldman, 1997*) theory of non-accidental features. Statistical relations that are unlikely to occur under a random model (e.g., collinearity of features) are evidence of the operation of non-random generative processes. Similarly, parts that have statistically distinct properties from the rest of the shape (*Schmidt et al., 2019*)—or which are statistically distinct from parts of objects seen previously (*Lake et al., 2015*)—are likely evidence of a category-defining process. Consistent with this idea, we find that parts that were deemed distinctive were also those that had the most significant effect of categorization (e.g., the 'twirl' feature in category 6; see *Figure 1a*). On a more global scale, curvature characteristics (straight or curved lines) of Exemplars were preserved almost perfectly in most drawings, showing that these features were also deemed non-random. Conversely, random features were altered more freely: Since the presented shapes were abstract, the part arrangement held no meaning and was therefore free to be modified (in contrast to familiar objects like a chair or a human), resulting in some Variations with changed part-ordering from the Exemplar. Yet an important open question is which properties of parts or features are used to determine their status as outliers, and how these properties are determined, based on familiar objects and processing constraints of the visual system. For example, even though distinctiveness is an intuitive concept—as demonstrated by the high agreement by observers in **Experiment 5**—it is challenging to define it formally. A large number of factors can affect the distinctiveness of a part or feature, ranging from purely geometric qualities such as frequency of curvature changes (*Baker et al., 2021*; *Attneave, 1957*; e.g., a tentacle vs. a stub) to comparative analyses within the object (difference between part and other object parts) or across other objects seen before (difference between part and other objects' parts). More generally, parts might be compared to an internal taxonomy of parts, each weighted with its probability of appearance, potentially informed by frequencies of parts in real-world objects or by their function (*Mukherjee et al., 2019*; *Tversky, 1989*). Another important open question is how parts are parameterized so that identity-preserving variations can be generated. Skeletal representations (e.g., *Feldman and Singh, 2006*) offer a promising avenue for potential representations (*Destler et al., 2019*; *Wilder et al., 2011*; *Wilder et al., 2018*; *Wilder et al., 2019*).

## Conclusions and future directions

Taken together, the results of our experiments suggest that humans are not merely passive observers, who assign objects to categories based on their relative similarities in a fixed feature space. Instead, a key aspect of human generalization is our ability to identify important signatures of generative processes and then to run internal routines that actively synthesize new objects that we have never actually observed. Thus, although drawing tasks present experimenters with challenges—particularly in terms of analyzing the resulting drawings—they also provide a promising avenue into research not only about object categorization but also human creativity. As seen in *Figure 1d*, there is a notable difference between participants as to how much they tended to deviate from the Exemplars, raising the question of what drives these individual differences. Another particularly fascinating open question is the extent to which the synthesis of novel objects by observers recruits physical simulation processes (*Battaglia et al., 2013*). It seems plausible that experience with the natural ways that objects, materials, animals, and plants move and change shape over time (*Schmidt and Fleming, 2016a*; *Spröte et al., 2016*; *Schmidt et al., 2016b*) might influence the types of variations that we tend to imagine (e.g., articulating limbs into different poses). The fact that observers can derive diverse but constrained variations from a single exemplar suggests a deep perceptual understanding about the ways things in the natural world tend to vary.

In future work, it would also be interesting to analyze the similarities between human drawings (and their similarities to the exemplars) using artificial neural networks trained on datasets of line drawings, such as the 'Quick Draw!' dataset (*Jongejan et al., 2016*; *Ha and Eck, 2017*; *Xu et al., 2021*; *Kabakus, 2020*). In particular, each of the human drawings (or exemplars) could be fed into such a network, and a feature vector describing the shape derived, allowing an automated quantitative similarity analysis to be performed. Here, we used human-based ratings to perform such analyses, but this approach does not scale well to larger datasets. Even more intriguing is the possibility of training a neural network model to reproduce the types of generalizations that humans produce when presented with a single exemplar. This would answer questions, such as whether it is necessary to be exposed to human-created line drawings in order to generalize like humans, or whether a visual diet

consisting entirely of natural images is sufficient. It would also allow a more rigorous test of whether part-based representations are necessary to capture the full range of human generalization. However, to date, one-shot generalization as exhibited by our observers remains a significant challenge for artificial vision systems.

## Materials and methods

Participants were recruited through university mailing lists. All participants gave informed consent before the experiments in accordance with the Declaration of Helsinki. The study was conducted in accordance with the Declaration of Helsinki and approved by the Ethics Committee of the Department of Psychology and Sports Sciences of the Justus-Liebig University Giessen (LEK-FB06; protocol code 2016-0007, approved 18 April 2016). **Experiment 1** was conducted on a touchpad computer, all others on a computer.

### Sample sizes

In the only studies known by the authors that employed a generative one-shot categorization task (*Feldman, 1997*), the number of objects drawn per Exemplar ranged from 96 to 168. Given that these Exemplars were comparatively simple (e.g., a dot on a line, two connected lines), we aimed for a substantially larger number of drawings per Exemplar (204) to allow all potentially relevant features of our Exemplars to be expressed in the generated objects. Given the novelty of this paradigm, estimating effect sizes for follow-up studies using the drawn shapes proved difficult. For this reason, all follow-up experiments in this study aimed for very high sample sizes to allow the detection of potentially very small effects. For example, in **Experiment 3**, in which chance level is at 12.5%, to detect a significant difference from that level of 2.5% using a one-sided binomial test (with Bonferroni-adjusted alpha = 0.00625 and power = 0.95) we would need a sample size of 3197 trials, a number we have exceeded (4800 trials) for additional sensitivity. Concerning *t*-tests, to detect a small Cohen's *d* of 0.2 (with alpha = 0.05 and power = 0.95), a minimum sample size of 1084 is necessary (for a one-sided *t*-test), which is exceeded, for example, in **Experiment 4** with 1200 data points per condition.

### Experiment 1: generative one-shot categorization task

Seventeen participants drew 12 new shapes for each of the 8 Exemplars (204 shapes per participant, overall 1632 shapes). They were instructed to draw a new object, belonging to the Exemplar's class, that was not just a mere copy of the Exemplar. The Exemplars were shown in randomized order in the upper part of an iPad Pro (12.9″) oriented in portrait mode and observers' drawings were on the lower part of the screen. While drawing, participants could switch between drawing freely and drawing straight lines. Participants could clear the current drawing or undo the last drawn segment. After putting down the pen they could only continue drawing from the end-point of the last segment to prevent gaps between lines. After finishing a shape, the drawing area was cleared and a new object could be drawn. A shape could only be finished if the contour was closed, that is, the first and last point drawn were on the same spot. Participants were instructed not to draw any overlapping or crossing lines.

### Experiment 2: perceived similarity to Exemplars

For each category, 12 new participants judged the perceived similarity of all drawings created in Experiment 1 to their originating Exemplar. Before making any judgements, all Variations of a category were shown in grids of 3 by 4 shapes consecutively to give the observers an idea of the range of shapes to be judged. In total, participants viewed 17 grids of 12 shapes. They were then presented with the Exemplar on the far left of the screen. On the screen bottom, participants used the mouse to drag and drop 3 randomly chosen Variations from that category into the upper part of the screen and arrange them based on similarity to the Exemplar: the closer the shape was placed on the *x*-axis towards the Exemplar, the more similar it was judged to be. Participants were instructed to try and keep a consistent scale of similarity, meaning that equally similar shapes relative to the Exemplar should be placed in the same area of the screen, across trials and irrespective of how similar the shapes were to each other. After placing three shapes relative to the Exemplar, a button press revealed the next three shapes on the screen's bottom portion. Already placed shapes were greyed

out but could still be adjusted in position if so desired. To prevent the screen from getting cluttered, old shapes disappeared after 4 trials so that only 12 shapes were shown at one time. For each of the participants responses, the similarity values were normalized between 0 (the Exemplar) and 1 (the least similar drawing). The final similarity value of each drawing was averaged across observers over these normalized responses.

### Picking stimuli from similarity space

In **Experiments 2b**, **3**, **4**, **5**, and **6**, we selected a subset of shapes from each category ensuring that the subset spanned the whole range of similarities. As an example, a subset of 20 shapes was created by taking all shapes of a category (minus the Exemplar) and dividing them into 20 equally sized bins spanning the perceived similarity space derived in **Experiment 2**. For each bin, we selected the shape with the lowest between-participant variance in similarity judgements. If a bin was empty, neighbouring bins were searched for a shape not yet used.

### Experiment 2b: comparing copies and new drawings

Fifteen participants were instructed to copy each Exemplar three times as best as possible with the drawing interface used in **Experiment 1**, resulting in 45 copies of each Exemplar. Then, 15 new participants were shown one of these copies, along with the Exemplar and one Variation of that category per trial. The task was to pick the shape that was a copy of the Exemplar. The 45 Variations per category were selected to span the range of similarities (see **Picking stimuli from similarity space**) and randomly paired with one of the copies. Each category comprised one block of trials with blocks being randomized in order, resulting in 45 × 8 = 360 trials. These trials were repeated three times resulting in 1080 trials overall. In each repeat, the order of blocks and pairings of copies and Variations was randomized.

### Experiment 3: perceptual category membership

Fifteen participants sequentially judged which Exemplar category 40 drawings of each category (chosen as described in **Picking stimuli from similarity space**) belonged to, resulting in 320 data points per participant. If participants were unsure about the category membership, they were instructed to pick the category they thought the shape belonged to the most.

### Experiment 4: corresponding parts experiment

Fifteen participants participated in this experiment. For each Exemplar, a subset of 10 drawings spanning the range of similarities was chosen (see **Picking stimuli from similarity space**). Participants were shown an Exemplar on the left and a drawing on the right, either belonging to the same or a different category. They were then asked if they saw any corresponding parts between these shapes. If not, the next shape pair was shown. Otherwise they were asked to pick the corresponding parts by first picking the part in the Exemplar and then the corresponding part in the drawing. To pick a part, the two delineating points of the part were to be clicked in succession, creating a line between these points within the shape. After that, the polygon on either side of the line could be chosen as the final part. A part could only be picked if the resulting line between the points did not intersect with other parts of the shape. Instead of picking a part, the rest of the shape (comprised of anything not yet picked) could also be picked, indicated by a red dot at the centroid of the remaining shape. Any section already picked could not be used for another part. Any number of part pairs could be picked, unless the remaining shape was picked, after which the trial was ended, since no more unpicked parts remained.

Each Exemplar was paired with 10 corresponding and 10 Variations of another randomly chosen category, resulting in 8 × 20 = 160 trials per participant.

### Part order analysis

To compare part ordering in Exemplar/Variation pairs we created a simplified representation of parts (see *Figure 3—figure supplement 2a*): Starting with the left-most point of a shape and moving in clockwise order, each point was tested whether it was within a designated part, or not. Accordingly, a value or an empty value (or 'gap') was added to an array. The array values only changed when encountering a new part (or non-part; i.e., if the current part was not the same as for the previous point). While this is a simplified representation of part order, we believe it to be a sensible approach for the

shapes of this study, given that they mainly consisted of one central main body with a differing number of parts attached. This analysis results in a circular array showing the clockwise ordering of parts for each shape ('part circle'). To compare part order, non-corresponding parts (i.e., gap segments) were ignored, since we were only interested in the order of corresponding parts. By comparing these arrays, we could see whether part ordering was identical, reversed, or shuffled. Since order between two circular arrays could only be different when they consisted of more than two values (i.e., the array [A,B] is considered the same as [B,A]), only trials with more than two correspondences were considered, amounting to 30% of trials. Part changes, for example, omitted or substituted parts of an Exemplar or parts added to the Variation that did not exist in the Exemplar, were defined by gaps in the part order array: A gap in the Exemplar with no counterpart in the Variation at that same spot was an omission. A gap in the Variation but not the Exemplar was considered an addition. If there was a gap at the same spot in both shapes it was defined as a substitution. For this part-change analysis all stimuli where any correspondence was seen was used (85% of trials).

## Experiment 5: marking distinctive parts

Ten participants were sequentially shown a subset of 39 shapes (chosen as described in **Picking stimuli from similarity space**) for each category, plus the Exemplar, resulting in 320 trials overall. They were asked to mark the 'most distinctive areas' of each shape. They could freely paint on the shape's silhouette and were not constrained to paint only consecutive parts. After painting the most distinctive part or parts in red they could switch to the next lower distinctiveness tier (orange) and paint the second most distinctive areas. After that they could switch to yellow for the third most distinctive area. At least one part had to be painted red, the lower distinctiveness tiers were optional.

To aggregate these responses (as in *Figure 4* and *Figure 3—figure supplement 1*), each point of the contour was given a score, with each red response adding 3, each orange 2 and each yellow adding 1 to the score of that point. After summing the individual scores, the resultant scores were normalized between 0 (score = 0) and 100 (highest possible score).

## Creation of randomized responses for comparison with data from Experiment 5

For each categories' responses, we computed how often each distinctiveness tier was used (except the first tier, which was always used). In addition, we computed the distribution of lengths of each tier's consecutive painted areas and the distribution of number of non-consecutive areas per tier and category. With these distributions 10 (the number of participants in **Experiment 5**) randomized responses were created for each of the shapes from **Experiment 5**, meaning the average number of areas and consecutive lengths mimicked the human distribution, while the placement on the shape was random.

## Experiment 6: swapping distinctive parts experiment. Stimuli

First, we segmented each shape of **Experiment 5** into one distinctive and one indistinctive part. The distinctive part was the largest consecutive part of the shape with an aggregated distinctiveness score higher than 75 in each point. The indistinctive part was the rest of the shape.

In 'distinctive part swapped' shapes, we swapped the distinctive part with a distinctive part of another categories' Exemplar. In 'indistinctive part swapped' shapes, we swapped an indistinctive part of the shape with the same number of points as the distinctive part that had the lowest average distinctiveness score. Since points were equally spaced along the contour both parts had the same perimeter length. In either case the swapped-in part was rotated and either uniformly compressed or stretched to fit the gap as best as possible.

In each case a drawing from the subset from **Experiment 5** (excluding the Exemplars) was used as the base shape. In this way for each of the 8 categories, 5 shapes were created with parts swapped from one of each of the other 7 categories both once with the distinctive part and once with the indistinctive part swapped, resulting in 8 × 7 × 2 × 5 = 560 shapes.

Because this process sometimes created shapes with large self-intersections, the final five shapes for each condition were hand-picked to create well-formed, artefact-free shapes.

## Procedure

Fifteen participants then conducted a repeat of the 'Perceptual category-membership' experiment with the 560 stimuli (see **Experiment 3**).

## Perceived Exemplar complexity

Twenty participants arranged the eight Exemplars in order of perceived complexity from simple to complex. The average rank of these responses was used to order the Exemplars and their categories in most plots of this paper.

## Acknowledgements

Research funded by the Deutsche Forschungsgemeinschaft (DFG, German Research Foundation—project number 222641018—SFB/TRR 135 TP C1), by the European Research Council (ERC) Consolidator Award 'SHAPE'—project number ERC-2015-CoG-682859 and by 'The Adaptive Mind', funded by the Excellence Program of the Hessian Ministry of Higher Education, Science, Research and Art.

## Additional information

### Funding

| Funder | Grant reference number | Author |
|---|---|---|
| Deutsche Forschungsgemeinschaft | 222641018-SFB/TRR 135 TP C1 | Roland W Fleming Filipp Schmidt |
| European Research Council | ERC-2015-CoG-682859 | Roland W Fleming |
| Hessian Ministry of Higher Education, Science, Research and Art | The Adaptive Mind | Roland W Fleming |

The funders had no role in study design, data collection, and interpretation, or the decision to submit the work for publication.

### Author contributions

Henning Tiedemann, Conceptualization, Data curation, Formal analysis, Investigation, Methodology, Resources, Software, Visualization, Writing - original draft; Yaniv Morgenstern, Filipp Schmidt, Conceptualization, Formal analysis, Methodology, Resources, Software, Supervision, Writing – review and editing; Roland W Fleming, Conceptualization, Formal analysis, Funding acquisition, Project administration, Resources, Software, Supervision, Writing – review and editing

### Author ORCIDs

Henning Tiedemann (iD) http://orcid.org/0000-0003-2637-7888
Yaniv Morgenstern (iD) http://orcid.org/0000-0001-6902-667X
Filipp Schmidt (iD) http://orcid.org/0000-0001-8023-7304
Roland W Fleming (iD) http://orcid.org/0000-0001-5033-5069

### Ethics

The study was conducted in accordance with the Declaration of Helsinki and approved by the Ethics Committee of the Department of Psychology and Sports Sciences of the Justus-Liebig University Giessen (LEK-FB06; protocol code 2016-0007, approved 18 April 2016).

### Decision letter and Author response

Decision letter https://doi.org/10.7554/eLife.75485.sa1
Author response https://doi.org/10.7554/eLife.75485.sa2

## Additional files

### Supplementary files
• Transparent reporting form

### Data availability

All data generated and analyzed are available at zenodo at: https://zenodo.org/record/5230306.

The following dataset was generated:

| Author(s) | Year | Dataset title | Dataset URL | Database and Identifier |
|---|---|---|---|---|
| Tiedemann H, Morgenstern Y, Schmidt F, Fleming RW | 2021 | One shot generalization in humans revealed through a drawing task - Dataset | https://zenodo.org/record/5230306 | Zenodo, 10.5281/zenodo.5230306 |

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
