## [Editor Report]

This paper employs innovative approaches to elegantly tackle the question of how we are able to learn an object category with just a single example, and what features we use to distinguish that category. Through a collection of rigorous experiments and analytical methods, the paper demonstrates people's impressive abilities at rapid category learning and highlights the important role of distinctive features for determining category membership. This paper and its approach will be of interest to those who study learning, memory, and perception, while also contributing to a growing field which uses naturalistic drawing as a window into high-level cognition.

---

## [Decision Letter]

**Decision letter after peer review:**

Thank you for submitting your article "One shot generalization in humans revealed through a drawing task" for consideration by *eLife*. Your article has been reviewed by 3 peer reviewers, and the evaluation has been overseen by a Reviewing Editor and Chris Baker as the Senior Editor. The following individual involved in review of your submission has agreed to reveal their identity: Wilma Bainbridge (Reviewer #2).

*Reviewer #1 (Recommendations for the authors):*

I really enjoyed the work, and my comments are predominantly ways I think the authors could strengthen their existing claims and better justify their conclusions. I think the experiments were generally well designed and the data interesting.

1. Associated with public review point 1 – how were the other strategies in Figure 7 defined? At one point, this excerpt is included: "a range of other, more complex strategies were also used to generate shapes that grouped with the Exemplar, and even some that did not", and it's not particularly clear what this means, or whether these other strategies were participant-identified, data-driven, or speculation on the part of the researchers.

2. Associated with public review point 1 – "At times the parts were shuffled, resulting in a shape with a different part ordering. Sometimes only a subset of the original parts was used, or parts were added with respect to the Exemplar." Is this backed by data? Potentially, you could take participant reports of overlapping features from Exp 4, then find the angular distance between the part's location on the Exemplar and its location on the Variation. Similarly, how often were parts omitted? How often were new parts added?

3. Associated with public review point 2 – I believe that Exp 3 and 4 fill some of this gap, but in E2, the similarity ratings do not tell us much about categorical membership, and to a certain extent, the finding that some Variations are rated more similar to the Exemplar than others is a necessary outcome of the task design. I believe a quick experiment that replicates this task but includes intermixed Variations from multiple exemplars would be useful, It would provide a more valid, continuous metric of similarity, and you would be able to test whether Variations are rated as more similar to their own Exemplar than another.

4. Associated with public review point 2 – The claim is made that "even when the part-structure was altered substantially, other observers were able to identify corresponding parts between Variations and Exemplar, with a high degree of consistency". I assume that when the part structure was altered substantially, these would also be the Variations rated as least similar to the Exemplar in E2. To make this claim then, would require an analysis of the low similarity Variations in particular. I imagine the authors have the data to establish this, but if not, the claim should be tempered or clarified. A similar claim is made in another section, that similarly requires empirical support or tempering: "…could be varied while still retaining its identity. Importantly, these assumptions seem to be shared by other observers."

5. Associated with public review point 3 – In E5, it is suggested that "these distinctive parts are a major driving force for correct categorization (Experiment 3)". This is something that could be explicitly tested via cross-experiment comparisons.

6. Associated with public review point 4 – How were the initial exemplars generated? Do we know anything about their overlap or similarity?

7. I was tripped up a bit by a few elements in the figures. In Figure 1, 'creativity' is probably a construct that is characterized by more than variance in drawings, so maybe it should just be labelled as something more precise like "Intra-exemplar Similarity". Similarly in Figure 2, these aren't predicted categories per se, but rather participant responses. In Figure 4, what are the numerical values that correspond to points on the color bar? In Figure 5, "Categorization equals category of swapped in part" could be subtly changed to something like "Categorization consistent with category of swapped in part".

8. It would be useful to see some discussion of what exactly defines an individual feature, or what makes a person categorize a feature as distinct. Could it be a function of the set of Exemplars? That is, a feature is distinct if it is not present in the other Exemplars? Is it some metric of deviation from a rounded shape?

9. This is not necessary for the current paper, but it may be worth feeding these shapes through a neural network pre-trained on line drawings (e.g. sketchRNN, or anything trained on "Quick, Draw"). Then, you could derive a feature vector for each and compute a RDM for all shapes, comparing across all categories.

---

## [Author Response]

Reviewer #1 (Recommendations for the authors):I really enjoyed the work, and my comments are predominantly ways I think the authors could strengthen their existing claims and better justify their conclusions. I think the experiments were generally well designed and the data interesting.1. Associated with public review point 1 – how were the other strategies in Figure 7 defined? At one point, this excerpt is included: "a range of other, more complex strategies were also used to generate shapes that grouped with the Exemplar, and even some that did not", and it's not particularly clear what this means, or whether these other strategies were participant-identified, data-driven, or speculation on the part of the researchers.

We have now significantly expanded and clarified the speculative nature of these additional strategies, and clarify that they might only be a subset of many potential strategies. See the new ‘Ideas and Speculations’ sub-section of the manuscript.

2. Associated with public review point 1 – "At times the parts were shuffled, resulting in a shape with a different part ordering. Sometimes only a subset of the original parts was used, or parts were added with respect to the Exemplar." Is this backed by data? Potentially, you could take participant reports of overlapping features from Exp 4, then find the angular distance between the part's location on the Exemplar and its location on the Variation. Similarly, how often were parts omitted? How often were new parts added?

As described above, we have added a new part-order analysis to Experiment 4 (also see Figure S 4) and discuss the its results and limitations in the Discussion section. Overall, most Variations analysed for part-order retained the part order of the Exemplar. Some Variations, however, had a shuffled order, with exact percentages per similarity to Exemplar shown in Figure S 4.

3. Associated with public review point 2 – I believe that Exp 3 and 4 fill some of this gap, but in E2, the similarity ratings do not tell us much about categorical membership, and to a certain extent, the finding that some Variations are rated more similar to the Exemplar than others is a necessary outcome of the task design. I believe a quick experiment that replicates this task but includes intermixed Variations from multiple exemplars would be useful, It would provide a more valid, continuous metric of similarity, and you would be able to test whether Variations are rated as more similar to their own Exemplar than another.

We agree that the suggested task would result in a more robust similarity metric, and especially the comparison across categories would be very useful. However, given how often Variations were classified correctly in Experiment 3, we believe that the proposed task would at the same time strongly reduce the number of shapes considered of the same category—resulting in a well-populated stimulus space close to the Exemplar, surrounded by a large, sparsely-populated gap, and more remote locations with all shapes from the other categories. Therefore, this task would be an effective way to filter out ‘unsuccessful’ Variations (which clearly do not seem to belong to the respective Exemplar), however, at the cost of resolution for the more similar shapes. Consequently, we believe that our intra-category similarity space combined with the inter-category discrimination of Experiment 3 does provide sufficient data to support our conclusions. However, we agree with the reviewer that the alternative task they propose would be a useful extension for follow-up studies that are focused on cross-category boundaries or confusions.

4. Associated with public review point 2 – The claim is made that "even when the part-structure was altered substantially, other observers were able to identify corresponding parts between Variations and Exemplar, with a high degree of consistency". I assume that when the part structure was altered substantially, these would also be the Variations rated as least similar to the Exemplar in E2. To make this claim then, would require an analysis of the low similarity Variations in particular. I imagine the authors have the data to establish this, but if not, the claim should be tempered or clarified. A similar claim is made in another section, that similarly requires empirical support or tempering: "…could be varied while still retaining its identity. Importantly, these assumptions seem to be shared by other observers."

We addressed these points by an extensive rewriting of this part of the discussion, now taking into account the results of the new part order analysis (Experiment 4, Figure S 4) as well as the analysis targeting the relationship between the percentage of corresponding area with similarity (Figure S 2) (line 462 onwards).

5. Associated with public review point 3 – In E5, it is suggested that "these distinctive parts are a major driving force for correct categorization (Experiment 3)". This is something that could be explicitly tested via cross-experiment comparisons.

While we agree that cross-experiment comparisons could potentially provide additional evidence to support this claim, we believe we already have stronger evidence than such analyses would provide. To assess the causal role of distinctive parts with cross-experiment comparisons would be challenging, and we believe less informative than what we have done. First, since errors in Experiment 3 were rare, we would have only few data points to analyse mis-categorizations. Second, any comparative analysis would only provide correlational rather than causal evidence. This is why, instead of performing cross-experiment comparisons, we conducted Experiment 6 in which we directly manipulate the presence or absence of distinctive parts. We believe this provides a more direct test of the causal role of distinctive parts in categorization.

6. How were the initial exemplars generated? Do we know anything about their overlap or similarity?

We added a more in-depth explanation how and why we hand-crafted the Exemplars to Experiment 1. It is not trivial to quantify shape similarity, however for the purpose of review, here we have included a very straightforward dissimilarity analysis based on the 1-“Intersection-over-Union” (IOU) metric that is widely used in computer vision. Values of 0 indicate identity (intersection and union of the two shapes are the same), values of 1 indicate no overlap between the shapes. As can be seen, most shapes are substantially different from one another, with 20 out of 28 cross-category comparisons having a dissimilarity value >=0.5. We take this as evidence that the exemplars were substantially distinct from one another, certainly far above discrimination thresholds. However, given the limitations of using IOU-based metrics to capture human shape-similarity, we decided not to include these more detailed quantitative evaluations of the shape in the manuscript.

**Author response image 1. sa2fig1:** 

7. I was tripped up a bit by a few elements in the figures. In Figure 1, 'creativity' is probably a construct that is characterized by more than variance in drawings, so maybe it should just be labelled as something more precise like "Intra-exemplar Similarity". Similarly in Figure 2, these aren't predicted categories per se, but rather participant responses. In Figure 4, what are the numerical values that correspond to points on the color bar? In Figure 5, "Categorization equals category of swapped in part" could be subtly changed to something like "Categorization consistent with category of swapped in part".

We agree and have added quotation marks to point out that we follow a particular definition of ‘creativity’ which is given in the figure captions of Figure 1. Also, we annotated the color bars in Figure 4 to clarify the range and changed the wording in Figure 5.

8. It would be useful to see some discussion of what exactly defines an individual feature, or what makes a person categorize a feature as distinct. Could it be a function of the set of Exemplars? That is, a feature is distinct if it is not present in the other Exemplars? Is it some metric of deviation from a rounded shape?

We have extended the Discussion section and now discussing the concept of distinctive parts as well as theories of what makes a part or feature distinctive in greater detail. See the new “Ideas and Speculation” sub-section.

9. This is not necessary for the current paper, but it may be worth feeding these shapes through a neural network pre-trained on line drawings (e.g. sketchRNN, or anything trained on "Quick, Draw"). Then, you could derive a feature vector for each and compute a RDM for all shapes, comparing across all categories.

This is a fantastic suggestion for a future paper and indeed, in collaboration with deep learning specialists, we are making some initial explorations in this direction. For the purposes of our claims in the current study, we believe that the human-based similarity ratings are sufficient, but it would certainly be exciting to automate the process. To address the comment, we have now added a paragraph discussing DNNs to the end of the manuscript (now called “Conclusions and future work”).